# The Current Role and Future Applications of Machine Perfusion in Liver Transplantation

**DOI:** 10.3390/bioengineering10050593

**Published:** 2023-05-15

**Authors:** Sebastian M. Staubli, Carlo D. L. Ceresa, Joerg M. Pollok

**Affiliations:** 1HPB and Liver Transplantation Service, Royal Free London NHS Foundation Trust, Pond Street, London NW3 QG, UK; 2Oxford University Hospitals NHS Foundation Trust, University of Oxford, Oxfordshire OX3 9DU, UK; 3Division of Surgery & Interventional Science, University College London, London WC1E 6BT, UK

**Keywords:** orthotopic liver transplantation, normothermic machine perfusion, hypothermic machine perfusion, normothermic regional perfusion

## Abstract

The relative paucity of donor livers suitable for transplantation has sparked innovations to preserve and recondition organs to expand the pool of transplantable organs. Currently, machine perfusion techniques have led to the improvement of the quality of marginal livers and to prolonged cold ischemia time and have allowed for the prediction of graft function through the analysis of the organ during perfusion, improving the rate of organ use. In the future, the implementation of organ modulation might expand the scope of machine perfusion beyond its current usage. The aim of this review was to provide an overview of the current clinical use of machine perfusion devices in liver transplantation and to provide a perspective for future clinical use, including therapeutic interventions in perfused donor liver grafts.

## 1. Introduction

Machine perfusion (MP) was first introduced in liver transplantation in the 1930s, when the first experiments were conducted to maintain the viability of human organs perfused ex vivo [1]. Subsequently, in the 1960s, perfusion systems using diluted blood under hyperbaric conditions were used in some of the first successful liver transplantations [2]. However, the main breakthrough was achieved shortly thereafter with the introduction of cold flush preservation solutions, known as static cold storage (SCS), which represents the current gold standard [3]. In SCS, the liver is perfused with several litres of 4 °C cold preservation fluid through the portal vein and hepatic artery before being packed and placed on ice for transport. Several solutions have been developed for this purpose, with the University of Wisconsin (UW) solution emerging as the gold standard owing to the addition of larger sugar molecules that stay in the extracellular space to maintain cellular osmolality [4].

Ischemia leads to the accumulation of metabolic products, causing ischemia/reperfusion injury. Upon reperfusion, an efflux of these metabolites, first, has a direct toxic effect and, second, triggers an inflammatory response, damaging hepatocytes in the graft organs [5]. The benefits of SCS compared with allograft preservation solely by placement on ice are its abilities to cool the organ from its core to maintain cellular viability and to decrease cellular swelling and rupture. Overall, SCS offers relevant advantages, such as simplicity and cost-effectiveness, and is adequate in low-risk organs. However, postoperative graft function decreases after 8 h in suboptimal organs and after 12 h in lower-risk donor organs under SCS conditions [6]. Four main weaknesses of SCS have been outlined in the past: ‘(I) sustained organ injury is not reversed; (II) further organ injury during storage continues; (III) organ viability cannot be assessed; and (IV) storage time is limited’ [7].

Recently, the demand for donor organs has steadily increased, and the need for suitable organs for transplantation is expected to further increase in the future [8]. In the United Kingdom, transplant activity and the number of listed patients decreased during the COVID-19 pandemic, but the number of listed patients increased by 424% since 2021 as a direct consequence from reactivating less-urgent patients post-COVID-19, reaching an all-time high. As a direct consequence, the median waiting time for transplantation has increased to 84 days (95% confidence interval [CI], 74–94 days), which leads to death or removal from the list due to a deteriorating condition in 8% of patients [9]. Thus, the number of deceased organ donors has almost doubled over the previous 10 years, mostly due to the expansion of the higher-risk donor population [9]. The current adiposity pandemic has contributed to this issue, as steatotic donor organs are becoming an increasing issue [10,11]. The main risk factors of delayed allograft function and organ loss include hepatic steatosis, donation after circulatory death (DCD), donor age, and prolonged cold ischaemia time (CIT) [12]. Conversely, standard-risk donor organs are from younger donors after brainstem death (DBD) and show superior outcomes compared with organs from DCD due to lower incidence rates of primary non-function, ischemic cholangiopathy (IC), graft loss, and death [13]. However, the reported outcomes of high-volume single-centre-experience DCD liver transplantation have shown comparable rates of graft and patient survival outcomes, with similar rates of IC between DBD and DCD liver transplant recipients [14]. These results have reignited the use of DCD livers worldwide.

In the context of organ shortage, the anticipated increase in the number of liver transplantations, and the use of DCD grafts, further expansion of the donor pool is necessary. MP allows for the use of higher-risk donor organs, increasing the number of transplantable organs. It might reduce CIT, especially in marginal organs, which are more susceptible to ischemia and reperfusion injury [15]. The two main forms of MP that are currently used are hypothermic (HMP) and normothermic MP (NMP). HMP relies on reduced cellular metabolism due to hypothermic conditions and the simultaneous washout of metabolites and toxins, whereas NMP allows for ongoing cellular metabolism at body temperature, allowing for viability assessment under near physical conditions [16]. Prolonged preservation time due to MP can help overcome logistical challenges, a frequent issue in clinical practice. Another added benefit of perfusion devices is the assessment of organ quality based on objective parameters, compared with ‘gut feeling’ or purely morphological features [17,18]. In the future, the evolution of MP devices from pure organ preservation and assessment systems towards the improvement and modulation of graft quality, such as the augmentation of lipid metabolism to decrease steatosis in the perfused liver, might further expand the available organ pool [19]. In this rapidly evolving field, the most effective, reliable, and clinically implementable perfusion procedure remains to be identified (Table 1).

## 2. Hypothermic MP

Hypothermic MP (HMP) is an umbrella term for various forms of MP that use hypothermic perfusion as their smallest common denominator. Systems that provide hypothermic oxygenated perfusion are termed hypothermic oxygenated perfusion (HOPE) if the perfusate is delivered via the portal vein and dual hypothermic oxygenated perfusion (D-HOPE) if the perfusate is additionally delivered through the hepatic artery.

The first prospective trial to evaluate ex situ hypothermic MP, which included 20 adult HMP livers with matched SCS livers, was published in 2010. The authors reported decreased early allograft dysfunction (EAD) rates (5% in HMP vs. 25% in the controls; *p* = 0.08), significantly lower values of serum injury markers, and a shorter hospital stay in the HMP group [20]. A further milestone was achieved in the first study to evaluate HOPE for DCD (*n* = 8) compared with DBD grafts (*n* = 8). All organs were treated with SCS, and DCD organs were treated with HOPE for 1–2 h before implantation. The postoperative outcomes were similar, showing the protective capacity of HOPE for DCD livers [21]. These results were confirmed by the same group with a larger cohort of 50 DCD and matched SCS livers in each group [22].

However, high-level evidence of the clinical value of HMP is relatively scarce, with only one published randomised controlled trial (RCT) comparing D-HOPE with SCS [23]. In this trial, 160 patients were randomised to an HMP or SCS group (78 patients in each group). Four patients did not receive an organ in this trial. The primary end point was symptomatic non-anastomotic biliary strictures 6 months after transplantation, and the secondary end points included intraoperative reperfusion syndrome and primary nonfunction. In the HMP group, the incidence of non-anastomotic biliary strictures was 6% compared to 18% in the control group (risk ratio [RR], 0.36; 95% CI, 0.14–0.94; *p* = 0.03). The incidence rates of postreperfusion syndrome and EAD were, respectively, 12% and 27% in the HMP group (RR, 0.43; 95% CI, 0.20–0.91) and 26% and 40% in the control group (RR, 0.61; 95% CI, 0.39–0.96). The reported findings of this trial showed a relatively short CIT of 6 h 11 min and low-risk donors. Thus, whether the effects would be replicable with higher-risk transplant organs remains to be elucidated.

A meta-analysis of nine studies, including the above-mentioned randomised trial and eight non-randomised trials, reported that HMP showed reduced incidence rates of EAD (OR, 0.51; 95% CI, 0.34–0.76; *p* = 0.001; *I*^2^ = 0%) and non-anastomotic biliary strictures (OR, 0.34; 95% CI, 0.17–0.67; *p* = 0.002; *I*^2^ = 0%) compared to SCS. Two trials used non-oxygenated HMP.

A recent study that could stimulate further research and the use of extended criteria donor (ECD) DBD livers was recently published. In this trial, ECD DBD organs were randomly assigned to either the HOPE or SCS group before implantation (*n* = 23 in each group). The primary outcome (peak ALT levels within 7 days after LT) was significantly decreased in the HOPE group. Secondary outcomes, including major complications and duration of ITU stay, were both significantly decreased in the livers randomised to HOPE. This trial also raised the question of ideal end points in MP trials, which is controversial [24]. Similar results were reported in an Italian trial that included ECD grafts from DBD donors randomly assigned to either HOPE (*n* = 66) or SCS (*n* = 69), showing a significantly lower EAD rate (13% vs. 35%, *p* = 0.007). However, 25 grafts were excluded after randomisation, making the results difficult to interpret [25]. A trial in the Netherlands that compares Normothermic Regional Perfusion (NRP) and DHOPE with controlled oxygenated rewarming NMP in ECD-DCD donation is currently recruiting (NCT05327478). Furthermore, a largely understudied field entails liver splitting under HOPE conditions, and a recently published pilot study demonstrated the feasibility of this approach with eight successfully transplanted split grafts under these conditions [26]. Further studies are necessary to elucidate the role of HOPE in split grafts.

In summary, hypothermic oxygenated MP has been demonstrated to lead to improved graft survival and decreased complication rates. This is likely due to the replenishing of cellular ATP reserves and the preparation of the organ for reperfusion under normothermic conditions during transplantation. Through the mechanism of cellular protection, the utilisation of marginal organs could be increased in the future to keep up with the increasing requirements of donor organs. It is our belief that the true value of hypothermic machine perfusion lies in the preservation of DCD grafts, as evidenced by the significant reduction in symptomatic non-anastomotic biliary strictures in D-HOPE livers in the RCT by the Groningen Group. Whilst there is some evidence to suggest the attenuation of reperfusion injury in DBD livers, this does not necessarily translate into a survival benefit (no study has been adequately powered to demonstrate improved survival). The true value of machine perfusion in marginal DBD grafts is likely to lie in a viability assessment, where the role of HOPE/D-HOPE is not yet well established. We consider this in more detail later in the manuscript (Table 2).

## 3. Normothermic MP

NMP entails the perfusion of a liver graft under normal body temperature with oxygenated packed red blood cells suspended in a colloid solution, nutrients, and medication, allowing the organ to maintain function under simulated physiological conditions. NMP devices are mainly composed of one or two pumps (portal venous and hepatic arterial flow), a blood reservoir, a heat exchanger, and an oxygenator. Different setups have been described in the literature [31]. NMP was first described in a porcine model in 2009, with positive outcomes in DBD and DCD livers, which were translated into a clinical trial in 2016, confirming the feasibility and safety of this approach [32,33]. Subsequently, a phase III multicentre RCT with 133 SCS and 137 NMP livers was conducted, reporting half the number of discarded livers in the NMP group compared to the SCS group (12% vs. 24%; *p* = 0.008), despite the significantly longer preservation time (714 min vs. 465 min; *p* < 0.001) [29]. In addition, parameters to predict graft quality were identified, including lactate clearance, pH maintenance, metabolism of glucose, and production of bile. In this trial, graft survival was similar between the groups, but decreased cellular damage in the NMP group with prolonged ex vivo organ preservation and increased organ utilisation without adverse outcomes were considered strong arguments in favour of NMP. A main point of criticism regarding NMP has been that, for logistical reasons, organs are usually treated with SCS until the receiving transplant centre performs NMP. In a recently published RCT, a portable NMP device was used to circumvent the use of SCS. In this trial, patients were randomly assigned to either a portable NMP (*n* = 153) or SCS group (*n* = 147) group. The primary outcome of the trial was EAD, which showed a significant reduction in the portable NMP group (18% vs. 31%, *p* = 0.01). Furthermore, the incidence of ischemia-reperfusion injury after reperfusion (defined as less moderate-to-severe lobular inflammation) was significantly reduced (6% vs. 13%, *p* = 0.004). The higher-use frequency of DCD livers, as previously reported by Nasralla et al., was confirmed in this study (51% vs. 26%, *p* = 0.007) [28].

A main argument in favour of NMP is that it allows for the measurement of objective parameters related to graft function and the identification of transplantable marginal organs that otherwise would have been discarded [34]. The identification and definition of the relevant parameters and their cut-offs are ongoing, but a recently published study that used 203 NMP perfused livers identified alanine transaminase and lactate after 2 h and peak bile pH and supplementary bicarbonate levels to keep the perfusate within 11 to 29 mmol at pH > 7.2 in the first 4 h as the strongest predictors of the outcome in a multivariate analysis [35]. The in-depth assessment of mitochondrial markers in 50 NMP livers revealed that the markers of mitochondrial integrity and viability correlated with clinical outcomes [36].

Conversely, a relevant shortcoming of NMP is its high cost compared to HMP/HOPE or SCS, although its cost-effectiveness has recently been demonstrated [37]. However, if NMP would gain further traction and its use become more widespread in the future, then market prices could be positively impacted. Furthermore, a combination of different perfusion methods has been attempted in the past, such as D-HOPE, controlled oxygenated rewarming, and NMP, but their roles and usefulness require further clarification [38]. This combined approach could really harness the individual benefits of each perfusion technology, and by initiating end-ischaemic D-HOPE (to protect the graft from ischaemia reperfusion injury) followed by NMP (for viability assessment), the optimal preservation of the liver could be achieved. The Groningen Group recently published results from their sequential hypothermic and normothermic machine perfusion protocol in high-risk donor livers. Fifty-four donor livers were preserved with a combination of D-HOPE and NMP, of which 34 were transplanted (63% utilisation). Importantly, post-transplant outcomes were excellent, with 94% one-year graft survival and cholangiopathy occurring in only 1 (3%) patient [39]. 

## 4. Normothermic Regional Perfusion

NRP has shown great promise in DCD liver transplantation and donor pool expansion [39]. To this end, its use is mandated for DCD organ donors in France. Its rationale is that prolonged warm ischemia leads to impaired graft function and graft loss, ultimately resulting in inferior clinical outcomes [40]. Prolonged warm ischemia times occur after the withdrawal of life-sustaining treatment and before organ retrieval. To mitigate ischemic damage to tissues and organs, the early iliac cannulation and perfusion of abdominal organs are performed shortly after donor circulatory arrest, re-establishing the perfusion of these organs with oxygenated blood [38]. This process allows for the early reversal of ischemic injury, replenishing cellular ATP stores, the assessment of graft viability, and ischemic preconditioning [41,42].

To date, no RCT involving NRP has been published, but retrospective series have shown superior outcomes after NRP transplantation compared to SCS transplantation [39]. These studies used a propensity score-matched comparison and concluded that biliary complications, including ischemic biliary type lesions and graft loss, were significantly less frequent in the NRP group. No direct comparison of NMP and NRP in the form of an RCT is available, but propensity-matched comparisons between the two modalities have shown comparable outcomes [40]. Owing to the significant heterogeneity and overall low lumbers of reported cases, a meta-analysis concluded that NRP in liver transplantation reduced postoperative biliary complication rates when compared to SCS, but further studies and high-level evidence are needed to elucidate the true clinical value of this approach.

## 5. Viability Assessment during MP

With the increasing adoption of MP strategies in clinical practice, defining a precise and important role for their use, compared to SCS, is becoming increasingly important. Given that any MP technology is more costly and logistically complex, justification for its use in clinical practice is required. Although clinical trials have demonstrated improved early outcomes, such as a reduction in EAD [43], the clinical relevance of this as an outcome has been questioned [44]. Furthermore, with 1-year survival rates after liver transplantation already at approximately 90%, any gains that can be achieved are likely to be marginal and not through one intervention alone. Other factors might also predict graft survival after transplantation beyond the liver preservation method. To this end, one main benefit of MP could be the increase in the number of livers transplanted. Increasing organ utilisation is important not only for patients on the transplant waiting list but also for a much bigger population: those with liver diseases who become too unwell for a transplant or those with liver disease who do not meet the current listing criteria (extended indications). MP could increase organ utilisation by de-risking transplants and allowing clinicians to obtain more objective information about liver function rather than base their decision to transplant the organ on putative donor characteristics.

NMP has been generally accepted as the optimal technology for assessing ex situ liver function [45]. This is mainly because a normo-thermically preserved liver exhibits many functional characteristics that are easily understood, interpreted, and contextualised by liver transplant surgeons. Early in the development of NMP, studies have identified the ability to predict liver viability based on parameters measured during perfusion. Parameters such as acid-based homeostasis, liver enzyme levels, hyaluronic acid clearance, factor V levels, and bile production correlated with survival in a porcine model of liver transplantation [46]. Given the functional complexity of the liver, which has more than 108 cells per gram weight and performs more than 500 functions, identifying and understanding the key perfusion markers deemed conducive to a successful transplant are challenging. Thus, it is likely that several parameters must be considered in combination to truly appreciate the liver’s function ex situ and performance after transplantation. Several groups have examined this in considerable detail, with the Birmingham team testing their viability criteria in a formal clinical trial [34]. Marginal livers, as defined by specific criteria, that were declined for transplantation by all UK transplant centres were assessed using the NMP. Viability was confirmed after 4 h of perfusion if the liver achieved lactate clearance to ≤2.5 mmol/L and two or more of the following criteria: bile production, evidence of glucose consumption, hepatic arterial flow rate of ≥150 mL/min and portal venous flow rate of ≥500 mL/ min, pH of ≥7.30, and presence of homogeneous perfusion. Thirty-one livers were assessed, and 22 were transplanted after meeting the pre-defined criteria. Although the trial met its primary end point and 100% 90-day graft survival was achieved, four livers developed ischaemic cholangiopathy (IC) that required re-transplantation. This study concluded that the criteria were sufficient to assess hepatocellular function but were not as useful in assessing cholangiocyte viability [34]. Another important conclusion from this trial was that not all livers can be salvaged through end-ischaemic NMP and that those that are more severely damaged may benefit from other perfusion modalities, such as NRP or continuous NMP.

Given the importance of protecting the biliary tree during preservation to prevent the catastrophic complication of IC in DCD livers, which can occur in up to 30% of cases, assessing cholangiocyte viability during NMP is critically important. Data from another trial showed that differences between the bile and perfusate levels of pH, bicarbonate, and glucose can be used to identify bile alkalisation and glucose reabsorption by the biliary epithelium, thereby providing biomarkers of bile duct integrity and viability [47]. When applying the criteria of bile production >10 mL and bile pH > 7.45, the Groningen group tested the viability of 16 livers, of which 11 were transplanted and none developed IC [38]. The Cambridge group also realised the importance of bile analysis. In their initial series of 47 perfused livers, 22 were transplanted, and among those that subsequently developed IC, the bile produced during NMP was less alkalotic (pH < 7.4) than those produced without biliary complications [48]. The histological assessment of these livers showed circumferential stromal necrosis in >50% of the septal bile ducts. Bicarbonate, an important constituent of bile, is secreted into the biliary canaliculi by cholangiocytes; failure to produce alkaline bile represents impaired bicarbonate secretion due to damage to the biliary epithelium, which later manifests clinically as IC. The same group recently published their experience from 203 perfused livers, representing the largest study on a viability assessment during NMP [35]. A total of 154 livers were transplanted based on the group’s already established viability criteria. A multivariable analysis revealed that 2 h ALT level, 2 h lactate level, and the 11–29 mL of sodium bicarbonate supplementation needed to achieve a perfusate pH of > 7.2 and peak bile pH were associated with early allograft function. Despite that all the livers were transplanted on the basis of bile glucose and pH assessments, 11% developed non-anastomotic biliary strictures [35]. This study highlighted that different liver qualities during NMP could be discriminated, but identifying livers that will fail after transplant remains problematic because the liver will only be transplanted after NMP, when it is believed to function. This subjects all the studies to considerable selection bias. To properly assess viability, a larger proportion of livers that would fail after transplantation must be included, but given the current state of the field, it would be difficult to justify transplanting a liver where post-transplant failure would be a major concern. Any analysis of machine-perfused livers that have already been selected as suitable for transplantation would unlikely define the viability criteria without a large cohort of failed grafts. However, a viability assessment could become more valuable for assessing the quality of grafts that could be transplantable but are likely to represent a spectrum of post-transplant functions. This can help transplant surgeons in identifying the most appropriate recipients for certain types of grafts, allocating a low-risk graft to a high-risk recipient and vice-versa. By adopting this tailored approach, organ utilisation can be maximised and can improve overall outcomes.

Cold preservation relies on the suppression of the metabolic rate, as most enzymatic reactions slow down with temperature reduction. Therefore, hypothermic perfusion technologies are traditionally considered less useful in assessing graft viability. Furthermore, in the cold, active bile secretion is low and is used as a viability parameter during NMP. Nevertheless, small-scale studies have suggested that perfusate transaminases, glucose, lactate dehydrogenase, and lactate levels correlate with early graft function [49,50,51]. Flavin mononucleotide (FMN) has emerged as a promising marker of mitochondrial complex I injury that can be measured in real time during perfusion. The assessment of FMN in the perfusate is predictive of severe graft dysfunction or early graft loss.

The Zurich group recently found that FMN, determined using fluorescence spectroscopy in HOPE perfusate, correlated with early graft loss, cumulative complications, and hospital stay after liver transplant [52]. If the FMN concentration increases to >8800 AU at 30 min of HOPE or sharply inclines, the authors recommended not transplanting the liver; for intermediate release (8800–5000 AU), they suggested allocating the liver to a low-risk recipient [52]. In a recent study by Patrono et al. [50], 10 consecutive livers were subjected to D-HOPE, and FMN appeared to accumulate in perfusate, with an incremental trend in grafts that developed EAD, with the highest levels observed only in the failed grafts [50]. FMN is a promising biomarker of liver viability, with a sound scientifically justified hypothesis. However, more data are required to provide further evidence to support its value in clinical practice.

## 6. Management of IC

One of the most important clinical outcomes of liver transplantation is non-anastomotic biliary strictures or IC. This is characterised by strictures, dilatation, or irregularity of intra- or extra-hepatic bile ducts proximal to the biliary anastomosis and results in significant morbidity and mortality with high graft loss rates. IC mostly affects DCD grafts, with an incidence of 15–30% [53]. Before the advent of MP, no interventions were reported to effectively reduce the risk of or treat IC, and centres tended to avoid transplanting DCD grafts as a result. Recently, NRP has emerged as a game changer in DCD liver transplantation. Commenced during the multi-organ retrieval procedure, NRP restores blood flow after cardiac arrest and is aimed at reversing warm ischaemic injury during the DCD process. It is maintained for around 2 h, after which cold perfusion and rapid retrieval are initiated. The United Kingdom recently reported their NRP experience from two centres [54]. Between 2011 and 2017, 43 liver transplants from 70 DCD-NRP donor retrievals were performed. The results were compared with those of 187 standard DCD liver transplants performed during the period. None of the NRP livers developed cholangiopathy compared to 27% in the standard DCD group (*p* < 0.0001). Furthermore, death-censored 5-year graft survival significantly improved in the NRP livers (*p* = 0.04) [15]. Spanish transplant centres have also been strong proponents of NRP and recently reported their experience [39]. Between 2012 and 2016, 152 DCD donors received NRP, resulting in 95 transplants. Post-transplant outcomes were compared with those in 117 standard DCD livers. IC was significantly reduced in the DCD-NRP livers, with an incidence of only 2% in this cohort (*p* = 0.008) [39]. France made NRP mandatory for all DCD liver retrievals and reported their experience from 123 transplanted NRP livers [55]. This corroborated the findings from the United Kingdom and Spain, where only one recipient (0.8%) developed IC. These livers were otherwise low-risk from young donors, and recipients were also carefully selected [17]. As with any perfusion-related activity, the effective use of the technology involves a learning curve. An NRP-associated liver attrition rate was reported, and in the UK experience, three livers were lost that might have otherwise been transplanted without NRP [54]. In the Spanish series, six technical NRP failures occurred, although five livers were still transplanted [39]. Eleven NRP-related losses were reported in the French cohort [55].

Before the data from the NRP studies emerged, hypothermic MP (HOPE and D-HOPE) was purported as the optimal preservation method to reduce the risk of IC. Schlegel et al. [56] compared 50 DCD liver transplants from livers subjected to HOPE with 50 DCD transplants from SCS livers. They demonstrated an insignificant reduction in IC between the groups (8% in HOPE vs. 22% in SCS, *p* = 0.09) and a high 5-year graft survival rate of 94% in the HOPE cohort compared to 78% in the SCS group (*p* = 0.024) when censored for tumour-related deaths [56]. However, this study had considerable flaws. First, the populations compared were from different centres in different countries and were poorly matched. Furthermore, 70% of the livers in the HOPE group compared to only 20% in the SCS group were transplanted for hepatocellular carcinoma (*p* < 0.0001). When reporting survival outcomes, the study excluded those with tumour-related deaths; thus, considering the large discrepancy in transplants for malignancy, this makes the tumour-censored survival data problematic. The Groningen group reported the results of a multicentre RCT that compared D-HOPE with SCS in DCD liver grafts, with the important primary end point of symptomatic non-anastomotic biliary strictures at 6 months [23]. A total of 156 livers were transplanted (78 in each group), and significantly fewer symptomatic non-anastomotic biliary strictures were observed in the D-HOPE group than in the SCS group (6% vs. 18%; RR, 0.36; 95% CI, 0.14–0.94; *p* = 0.03) [19]. These findings could have substantial impacts on DCD liver transplantation, as the rate of IC was reduced by D-HOPE. However, the donors included in this trial were otherwise low-risk and had short CITs (median, 6 h 11 min). Whether D-HOPE offers the same benefit for DCD grafts with longer functional warm ischaemia time and CIT, as might be required to salvage organs declined by other centres, is unclear.

Muller et al. compared the outcomes in 132 NRP livers (French cohort) and 93 HOPE livers (Swiss cohort) [57]. Although EAD was significantly reduced in the NRP group (20% vs. 68%, *p* < 0.001), no significant difference in IC was found between the groups, with 4.5% and 8.6% of patients developing IC in the NRP and HOPE group, respectively (*p* = 0.22) [57].

NMP has been considered to be less effective than NRP or (D)HOPE in protecting the biliary tree and, therefore, in reducing IC rates, as demonstrated in the studies by Mergental et al. [34] and Watson et al. [48], who reported IC rates of 30% and 25%, respectively. The former concluded that ‘it is clear that end-ischaemic NMP does not prevent the development of non-anastomotic biliary strictures in high-risk DCD organs’ [34].

However, these studies used NMP in the end-ischaemic setting, with a median CIT of 7.5 h before NMP. Furthermore, the livers transplanted in these studies were amongst the most marginal, having been declined by all UK liver transplant centres prior to recruitment. It could be that this period of cold ischaemia has been detrimental to the integrity of the bile ducts of these very high-risk grafts. In the study by Ceresa et al. [58], where end-ischaemic NMP was also adopted in more organs that met the standard criteria, no IC cases were reported. However, more recently, the Cambridge group compared outcomes in DCD livers treated with SCS, NRP, and NMP and demonstrated that, although both MP modalities resulted in significant improvements in post-transplant liver function compared to SCS, no NRP-treated liver developed IC compared to 14% of the SCS-treated livers (*p* = 0.001) and 11% of the NMP-treated livers (*p* = 0.009) [59]. The 3-year graft survival was higher in the NRP cohort (90%) than in both the SCS and NMP groups (76%) [59].

In the two published RCTs that compared NMP with SCS [28,29], a continuous NMP (or device-to-donor) approach was utilised. In this context, low IC rates were observed. In the first RCT [29] of 27 DCD NMP transplant recipients, only one developed IC (3.7%). In the PROTECT study [28], a significant reduction in IC was observed in the DCD NMP group compared to the SCS group at the 12-month follow-up (2.6% vs. 9.9%; *p* = 0.02). Mohkam et al. [40] conducted an international observational study to compare 68 NRP liver transplants with 34 continuous NMP liver transplants in DCD liver transplantation. They found comparable post-transplant outcomes, with no significant difference in IC rate (1.5% in NRP vs. 2.9% in NMP, *p* = 0.99), and similar 2-year graft survival rates (91.5% vs. 88.2%, *p* = 0.52).

These data suggest that NRP appears to be the current gold standard approach in DCD liver transplantation. HOPE/D-HOPE offers a more straightforward low-cost approach, which appears beneficial, but its applicability is limited, and more clinical DCD data are needed. In a device-to-donor setting (which is a logistical challenge), NMP appears beneficial for protecting the biliary epithelium, almost comparably with NRP. However, end-ischaemic NMP is unlikely to protect the biliary epithelium but allows for viability assessment.

## 7. Future Directions

MP has demonstrated potential usefulness in improving liver preservation, organ utilisation, functional assessment, and outcomes. However, one future perspective is the potential of liver-directed therapeutic interventions to further enhance marginal organs [60]. This has mostly been explored in the context of NMP, where a metabolically active organ can be treated to either reverse or modulate a disease process in the donor organ to better prime the organ for transplantation.

Liver steatosis (excess hepatocellular fat deposition) could be treated during ex situ normothermic perfusion. Consistent with the global obesity epidemic, steatosis is an increasingly common finding in organ donors [61]. Macrovesicular steatosis in > 30% of donors is an independent risk factor of primary non-function and EAD [62], and steatosis is the most common reason for discarding a liver after retrieval. NMP has been found to be useful for removing excess liver fat during preservation and, therefore, to reduce the risk of the transplanted organ. Jamieson et al. [63] used a porcine model of steatosis to examine the effects of NMP on liver fat. After 48 h of NMP, steatosis was reduced to 15% (from 28%), although no measurable change in liver function was observed. NMP was limited by the absence of any means of clearing fat from the perfusate [63]. To increase the mobilisation of fat from hepatocytes, a defatting cocktail was developed, which reduced the intracellular lipid content by >50% in just 3 h after its addition to the NMP of steatotic livers from obese Zucker rats [64]. Another study explored pharmacological defatting in a modified NMP circuit that incorporated l-carnitine and exendin-4 into a dialysis machine [65]. In two human livers treated with defatting agents, substantial releases of triglyceride and low-density lipoprotein with a 10% reduction in macrovesicular steatosis were observed [65]. A larger study randomly assigned 10 discarded steatotic human livers to either standard NMP or NMP supplemented with the defatting cocktail with l-carnitine that was used in the rat study [66]. In contrast to the non-supplemented livers, which did not show any histological change in steatosis, the organs treated with the defatting cocktail demonstrated a 40% reduction in steatosis on histological examination within 6 h after perfusion (*p* = 0.02). This was accompanied by increased lipid oxidation and exportation to the perfusate, enhanced mitochondrial function, decreased vascular resistance, and reduced values of the markers of hepatocellular injury and inflammation with improved biliary function [66]. Of the 10 livers in the study, only two had severe steatosis, and four showed either no or mild steatosis, suggesting that reduced hepatocellular lipid content is beneficial regardless of the severity of steatosis. Finally, a study involving discarded steatotic livers perfused 18 human livers with NMP for 48 h (*n* = 6), with the addition of a lipid apheresis filter (*n* = 6) or an apheresis filter with defatting interventions using l-carnitine, forskolin, and glucose and insulin reductions (*n* = 6) [67]. The apheresis filter reduced the perfusate triglycerides and cholesterol levels. The defatting agents increased fatty acid oxidation and concomitantly reduced steatosis, as measured on the basis of tissue triglyceride levels. These defatting strategies appeared applicable only in functional livers during NMP and do not restore function in metabolically inactive steatotic livers [67]. Although the findings suggest that defatting strategies improve cellular function during NMP, they do not transform non-viable livers into transplantable livers. Instead, these strategies, none of which have been tested in clinical practice, improve the quality of an already transplantable (according to NMP parameters) liver.

IC remains a major concern after the transplantation of DCD liver grafts. Biliary strictures are thought to be related to micro-thrombi in the peri-biliary plexus, assumed to form during hypoperfusion and circulatory arrest [68]. Kidneys undergoing NMP have shown the upregulation of fibrinogen gene expressions with ischaemia and the development of fibrin plugs in small vessels, which respond to thrombolytic treatment [69]. DCD livers that underwent NMP developed bile duct wall infarcts in association with fibrin plugs in the peribiliary plexus [48]. The Cambridge group explored this hypothesis in discarded human DCD livers treated with TPA and FFP (as a source of plasminogen) during NMP [69]. The post-perfusion histology of the treated livers revealed no fibrin plugs and stromal necrosis. Thereafter, the same group reported outcomes from nine livers transplanted after treatment with alteplase and FFP [70]. After a 6-month follow-up, one liver developed a hilar biliary stricture, and none of the treated livers had excessive bleeding upon reperfusion. This approach is promising, although several questions remain unanswered, specifically the optimal dosing and delivery strategies for alteplase and FFP.

The presented examples provide insight into the potential of NMP to further enhance liver grafts for transplantation through liver-directed therapeutic interventions. Another key area for current and future research application is the ability to prolong perfusion. While this may not be required in the context of routine clinical transplantation, prolonged preservation might be needed to allow time for a therapeutic intervention for liver reconditioning or repair. Eshmuminov et al. [71] first reported prolonged normothermic liver preservation. In their novel circuit, they identified the need for strict glucose regulation, dialysis, and control of haemolysis to achieve a 7-day preservation of porcine and human livers [71]. More recently, Lau et al. reported the successful ex situ normothermic preservation of a split human liver graft for 13 days [72]. A group from Sydney attributed their success to a dialysis filter for toxin removal and an air-oxygen gas blender for the intensive regulation of acid-base balance [73]. Thus, prolonged perfusion in the clinical context may soon become a reality.

## Figures and Tables

**Table 1 bioengineering-10-00593-t001:** Overview of contemporary clinical trials involving machine perfusion in liver transplantation.

Trial Identifier	Study Type	Intervention	Primary End Point	No. of Participants	Location	End Date
NCT02478151	Single-arm, prospective	NMP	PNF 90 days;	40	Canada	2023
			re-transplantation after 90 days;			
			recipient mortality after 90 days			
NCT04812054	RCT	HOPE	EAD	104	Poland	2024
		SCS				
NCT03456284	Single-arm, prospective	NMP	PNF and recipient mortality at 90 days	30	United States	2023
NCT04644744	RCT	HOPE	Postoperative complications (CCI)	213	Germany	2024
		NMP				
		SCS				
NCT05045794	RCT	SCS + HOPE	EAD	244	United States	2024
		SCS				
NCT05574361	Single-arm, prospective	HOPE	EAD	120	United States	2023
NCT03484455	RCT	HOPE	EAD	142	United States	2022 *
		SCS				
NCT04483102	Single-arm, prospective	NMP	Graft failure at 6 months, total number of patients treated (declined livers)	25	United States	2023
NCT04023773	Single-arm, prospective	HOPE + NMP	1-month recipient and graft survival	15	United States	2024
NCT02775162	RCT	NMP	EAD	267	United States	2021 *
		SCS				
NCT04862156	Single-arm, prospective	NMP	EAD	105	United States	2024
NCT03929523	RCT	End ischemic HOPE	EAD	266	France	2023
ISRCTN14957538	RCT	NMP	Transplanted livers	60	United Kingdom	2024
		NMP + defatting				
ISRCTN11552402	Prospective, observational	NMP	Transplanted livers	3264	United Kingdom, international	2026
ISRCTN36453355	RCT	NMP	Transcriptome	250	United Kingdom	2025
ISRCTN15211703	Single-arm, prospective	NMP + thrombolytic treatment	Post-reperfusion blood loss	60	United Kingdom	2023

Abbreviations: RCT, randomised controlled trial; HOPE, hypothermic oxygenated perfusion; SCS, static cold storage; EAD, early allograft dysfunction; PNF, primary non-function; NMP, normothermic machine perfusion. * Completed trial.

**Table 2 bioengineering-10-00593-t002:** Overview of published RCTs on machine perfusion in liver transplantation.

Intervention	Primary End Point	Results	No. of Participants	Comments	References
HOPE	Non-anastomotic biliary strictures at 6 months	HOPE, 6% strictures; SCS, 18% strictures (RR, 0.36; 95% CI, 0.14–0.94; *p* = 0.03)	160 (78; 78), 4 no liver	Post-reperfusion, 12% vs. 27%; EAD, 26% vs. 40%	[23]
SCS					
HOPE	Peak ALT levels	47% decrease in serum peak ALT (*p* = 0.030)	46 (23, 23)	ECD livers	[27]
SCS					
Portable NMP	EAD	EAD (27/150 [18%] vs. 44/141 [31%]; *p* = 0.01)	293 (151; 142)		[28]
SCS					
NMP	Peak ALT	50% decrease in NMP compared with SCS	(137, 133)	50% lower discharge rate in NMP	[29]
SCS					
NMP	Patient and graft survival after 6 months	1 death in the SCS group	20 (10,10)		[30]
SCS					

Abbreviations: EAD, early allograft dysfunction; ECD, extended criteria donation; HOPE, hypothermic oxygenated machine perfusion; SCS, static cold storage; NMP, normothermic machine perfusion; ALT, alanine aminotransferase; RR, risk ratio; CI, confidence interval.

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
