# Peer review of "The Current Role and Future Applications of Machine Perfusion in Liver Transplantation"

_bioengineering, 2023, doi:10.3390/bioengineering10050593_

Round 1

Reviewer 1 Report

The manuscript entitled “The Current Role and Future Applications of Machine Perfusion in Liver Transplantation”, is a review paper that refers to the current knowledge about liver storage and the place of the liver machine perfusion before transplantation in the deceased model.

Liver machine perfusion (LMP) is an interesting and “hot” strategy for resuscitation organs with serious injury, mainly for livers from expanded criteria donor (ECD) and donor after circulatory death (DCD). It is also a model for organ evaluation before liver transplantation and the platform for ex-vivo organ treatment in the further perspective. Despite many trials and reports, it is still unclear what storage strategy should be used depending what type of donor is explored.

Minor comments:

·      I was wondering, what is the author opinion about the role and application of HOPE and D-HOPE LMP.

·      The authors wrote: “Furthermore, a combination of different perfusion methods has been attempted in the past, such as DHOPE, controlled oxygenated rewarming, and NMP, but their roles and usefulness require further clarification [36]”. In my opinion the HMP-COR-NMP model deserve for the more detail description. Could you please refer to the paper below?

van Leeuwen OB, Bodewes SB, Lantinga VA, et al. Sequential hypothermic and normothermic machine perfusion enables safe transplantation of high-risk donor livers. Am J Transplant. 2022;n/a(n/a). doi:10.1111/ajt.17022

·      The authors wrote about the use of normothermic regional perfusion (NRP). There are countries in Europe where the NRP is obligatory for DCD donors (cat. III Maastricht). Could you please refer to this?

Author Response

I was wondering, what is the author opinion about the role and application of HOPE and D- HOPE LMP.

Many thanks for this comment. We have now included a further paragraph at the end of the hypothermic machine perfusion section (section 2) to provide some personal insights into this technology.

The authors wrote: “Furthermore, a combination of different perfusion methods has been attempted in the past, such as DHOPE, controlled oxygenated rewarming, and NMP, but their roles and usefulness require further clarification [36]”. In my opinion the HMP-COR-NMP model deserve for the more detail description. Could you please refer to the paper below

van Leeuwen OB, Bodewes SB, Lantinga VA, et al. Sequential hypothermic and normothermic machine perfusion enables safe transplantation of high-risk donor livers. Am J Transplant. 2022;n/a(n/a). doi:10.1111/ajt.17022

Thank you for identifying this shortcoming in our manuscript. We have now added a further paragraph in which we highlight the potential for sequential perfusion strategies as well as citing this important paper.

The authors wrote about the use of normothermic regional perfusion (NRP). There are countries in Europe where the NRP is obligatory for DCD donors (cat. III Maastricht). Could you please refer to this?

We have now included this in our manuscript, thank you.

Reviewer 2 Report

This is a review on the hypothermic- normothermic machine perfusion of the liver graft after harvesting and normothermic regional perfusion during harvesting.

I have some comments.

1.        (L133) NRP will be NRP (normothermic regional perfusion). In L191, NRP (normothermic regional perfusion) will be NRP.

2.        (L186) DHOPE will be D-HOPE.

3.        (L401) What is the background that end-ischaemic NMP will not prevent IC?

Author Response

(L133) NRP will be NRP (normothermic regional perfusion). In L191, NRP (normothermic regional perfusion) will be NRP.

Thank you for highlighting this; it has been corrected accordingly.

(L186) DHOPE will be D-HOPE.

Thank you for highlighting this; it has been corrected accordingly.

(L401) What is the background that end-ischaemic NMP will not prevent IC?

In the previous sentence we have cited the conclusion from the Birmingham VITTAL study where they hypothesize that end-ischaemic NMP does not protect the biliary tree in the most marginal of grafts. We have added another sentence to hopefully make this clearer.

Reviewer 3 Report

This work sums up recent literature about several machine perfusion techniques available clinical trials are listed in the paper. 

Minor comment:

Lines 124-131: please clarify the statement. It is not clear, from the text in its present form, which group’s outcome was superior.

Author Response

Lines 124-131: please clarify the statement. It is not clear, from the text in its present form, which group’s outcome was superior.

Thank you very much for highlighting this. We have now rectified the manuscript accordingly.

Round 2

Reviewer 1 Report

Congratulations and thank you for taking into consideration my suggestions.